# Diagnostic Accuracy of the LabTurbo QuadAIO Common Flu Assay for Detecting Influenza A Virus, Influenza B Virus, RSV, and SARS-CoV-2

**DOI:** 10.3390/diagnostics14192200

**Published:** 2024-10-02

**Authors:** Chi-Sheng Tai, Ming-Jr Jian, Tai-Han Lin, Hsing-Yi Chung, Chih-Kai Chang, Cherng-Lih Perng, Po-Shiuan Hsieh, Hung-Sheng Shang

**Affiliations:** 1Graduate Institute of Medical Science, National Defense Medical Center, Taipei 114, Taiwan; eric7142@gmail.com (C.-S.T.); cindyft12@gmail.com (H.-Y.C.); 2Division of Clinical Pathology, Department of Pathology, National Defense Medical Center, Tri-Service General Hospital, Taipei 114, Taiwan; mj0106@gmail.com (M.-J.J.); garlicbun84@gmail.com (T.-H.L.); mblkaiser@gmail.com (C.-K.C.); ponchenli@gmail.com (C.-L.P.)

**Keywords:** SARS-CoV-2, Influenza A, Influenza B, RSV, diagnostic accuracy, LabTurbo QuadAIO

## Abstract

**Background:** The coronavirus disease 2019 (COVID-19) pandemic has highlighted the urgent need for rapid and accurate diagnostic tools for upper respiratory tract infections (URTIs). Nucleic acid amplification tests (NAATs) have transformed URTI diagnostics by enabling the rapid detection of multiple pathogens simultaneously, thereby improving patient management and infection control. This study aimed to evaluate the diagnostic accuracy of the LabTurbo QuadAIO Common Flu Assay compared to that of the Xpert Xpress CoV-2/Flu/RSV Plus Assay for detecting SARS-CoV-2, Influenza A, Influenza B, and respiratory syncytial virus (RSV). **Methods:** A retrospective diagnostic accuracy study was conducted using nasopharyngeal samples from patients. Samples were tested using the LabTurbo QuadAIO Common Flu Assay and the comparator Xpert Xpress CoV-2/Flu/RSV Plus Assay. Positive and negative percent agreements (PPA and NPA) were calculated. **Results:** The LabTurbo Assay demonstrated a PPA of 100% and an NPA of 100% for SARS-CoV-2, Influenza A, and Influenza B, whereas it showed a PPA of 100% and an NPA of 98.3% for RSV. **Conclusions:** The LabTurbo QuadAIO Assay exhibited high diagnostic accuracy for detecting multiple respiratory pathogens, including SARS-CoV-2, Influenza A, Influenza B, and RSV. Despite the slight discrepancy in the NPA for RSV, the overall performance of the LabTurbo Assay supports its integration into routine diagnostic workflows to enhance patient management and infection control.

## 1. Introduction

The COVID-19 pandemic has dramatically reshaped the landscape of point-of-care testing (POCT) for the diagnosis of upper respiratory tract infections (URTIs). In the aftermath of the pandemic, substantial advancements in POCT technologies have significantly enhanced the rapid and effective diagnosis of URTIs [1,2]. These improvements are crucial for timely patient management and for curbing the spread of the virus [3]. As POCT evolves, particularly through antigen and molecular testing, it addresses the critical challenges posed by SARS-CoV-2 and other respiratory pathogens, markedly affecting clinical settings, especially during peak respiratory disease seasons. Misdiagnosis or delayed diagnosis can increase the risk of transmission, leading to suboptimal patient management. Previous research has emphasized the importance of a robust global health infrastructure featuring advanced surveillance systems, improved diagnostic technologies, and heightened international cooperation to manage both emerging and re-emerging respiratory diseases [4,5].

Recent innovations in antigen-based POCT have proven effective in detecting high viral loads of SARS-CoV-2, thereby facilitating the rapid identification of infected individuals, which is a critical step in controlling transmission rates [6]. Moreover, these advancements have expanded the reach of diagnostics beyond traditional centralized laboratories, providing faster and more accessible testing solutions that are crucial to the current health crisis [7]. Moreover, the introduction of nucleic acid amplification tests (NAATs) in POCT has revolutionized URTI diagnostics by enabling rapid, sensitive, and specific detection of pathogens directly at the site of patient care. NAAT-based POCTs, including multiplex platforms, can identify a wide range of respiratory pathogens within hours, a significant improvement over the lengthy turnaround times of traditional culture-based methods. Studies have shown that NAATs not only enhance diagnostic accuracy but also improve patient outcomes by enabling targeted treatments [8,9].

The ability to identify key respiratory viruses swiftly and accurately, such as SARS-CoV-2, Influenza A/B, and respiratory syncytial virus (RSV), is essential for the effective clinical management of infectious diseases, especially during seasons when these viruses overlap [10,11,12]. Moreover, as the pandemic subsided and other respiratory infections became more common, the ability to distinguish between typical respiratory viruses quickly (Influenza A (Flu A), Influenza B (Flu B), and respiratory syncytial virus (RSV)) and SARS-CoV-2 has become crucial, given the similarity in their symptoms [13,14,15]. The development of high-throughput NAATs has facilitated the simultaneous detection and differentiation of these pathogens in a single test, which is crucial for guiding treatment strategies and implementing infection control measures [16,17]. The integration of these rapid testing methods into clinical practice has significantly bolstered the diagnostic capabilities for URTIs in the post-COVID-19 era, ensuring more efficient patient management and optimized use of healthcare resources.

The availability of rapid and accurate diagnostic tools is indispensable for ongoing global efforts to manage respiratory pathogens effectively. The development of the LabTurb QuadAIO all-in-one molecular detection platform exemplifies state-of-the-art molecular diagnostic technology, utilizing advanced multiplex PCR techniques to simultaneously detect and differentiate RNA from multiple viruses. The operational procedures for both the LabTurbo QuadAIO Common Flu Assay and the Xpert Xpress CoV-2/Flu/RSV Plus Assay (Cepheid, Sunnyvale, CA, USA) are analogous. Both systems perform fully automated detection of COVID-19, Influenza A, Influenza B, and respiratory syncytial virus (RSV), from sample processing to result reporting. The LabTurbo QuadAIO Common Flu Assay (LabTurbo, New Taipei City, Taiwan) offers a potential solution by reducing costs by 50% compared with the Xpert Xpress CoV-2/Flu/RSV Plus Assay. This reduction could make PCR-based flu testing more affordable for healthcare facilities and the general public. 

In our clinical evaluation, we conducted a retrospective analysis of nasopharyngeal samples, testing them with the LabTurbo QuadAIO Common Flu Assay (LabTurbo, New Taipei city, China) against the comparator method, Xpert Xpress CoV-2/Flu/RSV Plus Assay (Cepheid Sunnyvale, CA, USA) to validate the clinical test results.

## 2. Materials and Methods

### 2.1. Study Design and Clinical Specimens

For the clinical evaluation study, patients were recruited from both the inpatient and outpatient departments at the medical center, Taipei City. Retrospective nasopharyngeal samples from patients, both positive and negative for SARS-CoV-2, Influenza A, Influenza B, and RSV, were analyzed (*n* = 276 positive samples = 157, negative samples = 119). These samples were tested using the LabTurbo QuadAIO Common Flu Assay and the selected comparator method, the Xpert Xpress CoV-2/Flu/RSV Plus Assay (US FDA EUA210505), following standard practice in our clinical diagnostic laboratories to confirm the clinical test results. Residual nasopharyngeal swab specimens with viral transport medium (VTM) were stored at −80 °C. This study was approved by the Institutional Review Board (TSGH IRB No. B202305091, registered on 19 July 2023). Informed consent was obtained from all participants involved in this study.

### 2.2. LabTurbo QuadAIO Common Flu Assay

The LabTurbo QuadAIO Common Flu Assay (LabTurbo, New Taipei City, Taiwan) offers innovative point-of-care molecular diagnostics for rapid, precise, and cost-effective flu pathogen detection (Figure 1A). The LabTurbo AIO system integrates silica membrane column-based nucleic acid extraction, real-time reverse-transcription PCR, and result interpretation.

Respiratory pathogens in nasopharyngeal swab samples collected in viral transportation medium (VTM) are tested for COVID-19, Flu A, Flu B, and RSV. Each test uses 300 μL of VTM. Pre-loaded extraction and RT-qPCR reagents in sealed LabTurbo AIO cartridges enable fully automated pathogen nucleic acid detection. After transferring 300 μL of specimen VTM to the designated well, the cartridge is loaded into the system for processing.

The target genes of the LabTurbo QuadAIO Common Flu Assay included RdRP (SARS-CoV-2), M (Flu A), M (Flu B), and N (RSV) genes. RNase P (RP) was used as an endogenous IC (Internal Control) to validate the cells collected from individual samples.

### 2.3. Xpert Xpress CoV-2/Flu/RSV Plus 

The Xpert Xpress CoV-2/Flu/RSV plus system (Cepheid, Sunnyvale, CA, USA) detects co-circulating respiratory viruses (Figure 1B). A 300 μL specimen is loaded into a specific test cartridge, sealed, and inserted into the GeneXpert instrument. Each cartridge includes a sample processing control (SPC) and probe check control (PCC). GeneXpert software automatically generates qualitative results and records cycle threshold (Ct) values for viral RNA quantification.

Xpert Xpress CoV-2/Flu/RSV Plus targets genes N2, E, RdRP (SARS-CoV-2), M, PB2, PA (Flu A), M, and NS (Flu B) and nucleocapsids A and B (RSV). The 45-cycle test, including extraction and amplification, takes 36 min. When comparing the performance, the Xpert Xpress CoV-2/Flu/RSV Plus system was used as the standard method.

### 2.4. Diagnostic Accuracy and Agreement of the LabTurbo QuadAIO Common Flu Assay in Comparison with Xpert Xpress CoV-2/Flu/RSV Plus

The remaining specimens were tested using both the LabTurbo QuadAIO Common Flu Assay and Xpert Xpress CoV-2/Flu/RSV Plus (Cepheid, Sunnyvale, CA, USA), depending on the residual quantity. All experimental procedures were performed according to the manufacturers’ recommendations. Cohen’s κ was calculated to measure the overall agreement between the LabTurbo QuadAIO Common Flu Assay and Xpert Xpress CoV-2/Flu/RSV Plus. The positive and negative percent agreements (PPA and NPA) of each pathogen target were calculated using the following formula: PPA = TP/(TP + FN) × 100% − NPA = TN/(TN + FP) × 100%, where FN, FP, TN, and TP represent the numbers of false negatives, false positives, true negatives, and true positives, respectively. The correlation between the threshold cycle (Ct) values for the LabTurbo QuadAIO Common Flu Assay and Xpert Xpress CoV-2/Flu/RSV Plus Assay was assessed by considering the R-squared value of the respective linear regression. Descriptive test statistics were calculated for the LabTurbo Quad AIO Common Flu Assay. 

## 3. Results

### 3.1. Clinical Performance of the LabTurbo QuadAIO Common Flu Assay in Detecting Respiratory Viruses

Clinical evaluation of the LabTurbo QuadAIO Common Flu Assay demonstrated exceptional performance in detecting multiple respiratory viruses. For SARS-CoV-2, the positive percentage agreement (PPA) was 100%. The negative percent agreement (NPA) was 100%. Similarly, the Influenza A assay and the Influenza B assay achieved a PPA of 100% and an NPA of 100%. For RSV, the assay reached a PPA of 100% and an NPA of 98.3%. These high agreement rates confirm the reliability and accuracy of the LabTurbo QuadAIO Common Flu Assay, making it an invaluable tool for detecting these respiratory viruses in clinical diagnostics (Table 1).

### 3.2. SARS-CoV-2 Detection

The LabTurbo Assay detected all 48 positive SARS-CoV-2 cases identified using the comparator method (Table 2). The LabTurbo Assay correctly identified all 119 negative cases, ensuring that there were no false positives or false negatives. The detailed results are presented in Table 2, which shows the individual sample IDs and cycle threshold (Ct) values obtained using these two methods.

### 3.3. Influenza A Detection

The LabTurbo Assay detected all 40 Influenza A virus-positive cases identified using the comparator method (Table 3). A comparison of the Ct values from the LabTurbo and Xpert methods for detecting influenza showed a high level of agreement and consistency, indicating that both methods are reliable for clinical diagnosis. Overall, both methods appeared to be effective for the detection of influenza in the samples. 

### 3.4. Influenza B Detection

The LabTurbo method demonstrated exceptional effectiveness in diagnosing Influenza B, as evidenced by its perfect agreement with the Xpert method on a comprehensive dataset. In our study, which included 30 positive and 119 negative cases, LabTurbo achieved a positive percent agreement (PPA) of 100% and negative percent agreement (NPA) of 100%. The above results indicated that the LabTurbo method has excellent sensitivity for successfully and accurately identifying all positive and negative cases in Influenza B virus detection (Table 1 and Table 4).

### 3.5. RSV Detection

The comparison of the LabTurbo QuadAIO and Xpert Xpress Assays for RSV detection revealed excellent diagnostic performance, with high agreement rates between the two methods (Table 5 and Appendix A). The NPA for RSV detection was 98.3%. This indicates that 117 of the 119 negative cases were correctly identified by the LabTurbo QuadAIO Assay. The two false positive results suggest a slightly higher tendency to detect RSV where it is not present compared with the Xpert Xpress Assay. Two false positive results were observed in the 119 negative cases, resulting in a false positive rate of approximately 1.68%. Despite this minor limitation, the LabTurbo QuadAIO Assay remains a highly effective tool for rapid and accurate detection of RSV in clinical settings.

### 3.6. Inter-Assay Correlation of Ct Values

Figure 2 shows the correlation between Ct values for the CoV-2/Flu/RSV Plus and LabTurbo QuadAIO Assays. The data points were generated from clinical specimens. The R square values for SARS-CoV-2, Influenza A, Influenza B, and RSV were 0.9912, 0.9258, 0.9932, and 0.9819, respectively. The average Ct value differences were as follows: for SARS-CoV-2, the difference was 1.71 ± 1.88 (−6.68 to 7.04); for Influenza A virus, the difference was 0.35 ± 1.22 (−2.45 to 3.14); for Influenza B virus, the difference was 0.01 ± 0.31 (−0.71 to 0.69); and for RSV, the difference was 0.99 ± 0.82 (−1.55 to 2.56).

## 4. Discussion

This study assessed the diagnostic accuracy of the LabTurbo QuadAIO Common Flu Assay against the Xpert Xpress CoV-2/Flu/RSV Plus Assay. The LabTurbo Assay demonstrated high positive and negative agreement rates for targeted respiratory viruses, confirming its sensitivity. For SARS-CoV-2, Influenza A, and Influenza B, both positive percent agreement (PPA) and negative percent agreement (NPA) were 100%. RSV showed 100% PPA and 98.3% NPA.

Nucleic acid amplification tests (NAATs) have revolutionized the detection and management of upper respiratory tract infections (URTIs) [18,19,20]. These advanced molecular diagnostic tools have become indispensable because of their ability to rapidly and accurately identify multiple pathogens simultaneously, thereby providing timely information that is crucial for effective patient management and infection control. 

The rapidity of multiplex NAATs in diagnosing respiratory infections cannot be overstated. Traditional diagnostic methods such as culture-based techniques often require several days to yield results, which can delay critical treatment decisions [21,22]. In contrast, multiplex NAATs, such as the LabTurbo QuadAIO Common Flu Assay, provide results within 1.5 h. This swift turnaround is particularly vital during peak respiratory illness seasons when timely differentiation among pathogens, such as SARS-CoV-2, influenza A/B, and RSV, can significantly influence clinical decisions and public health responses. 

Multiplex NAATs for detecting multiple pathogens in a single test streamline laboratory workflow and reduce healthcare facility burden. This approach minimizes the need for multiple tests, enhancing efficiency in clinical environments, particularly during peak respiratory infection seasons.

Rapid, accurate diagnosis enables timely, appropriate treatment, distinguishing between bacterial and viral infections to reduce unnecessary antibiotic use and guide targeted therapies. During the COVID-19 pandemic, the swift identification of SARS-CoV-2 and other respiratory viruses has been crucial for managing co-infections and resource allocation.

The Xpress CoV-2/Flu/RSV Plus Assay, priced at approximately USD 68 per test with a USD 68,000 system cost, proves prohibitive for routine flu testing. This high cost has impeded widespread PCR test adoption for common flu, limiting general population accessibility. The LabTurbo QuadAIO Common Flu Assay offers a potential solution with less than 50% costs. This reduction could enhance affordability of PCR-based URTIs testing for healthcare facilities and the public. 

Although the results of this study are promising, there are some limitations. The retrospective nature of this study meant that the samples were not collected prospectively, which could have introduced a selection bias. Additionally, this study was conducted at a single hospital and the findings may not be generalizable to other settings with different patient populations and RV prevalence rates of respiratory viruses. Further multicenter studies are needed to confirm these findings and establish the utility of the assay across diverse clinical environments. Future studies should focus on expanding the evaluation of the LabTurbo Quad AIO Assay to include a broader range of clinical settings and patient populations. This would help to verify the performance of the assay and ensure its generalizability. Additionally, it would be beneficial to explore the effectiveness of this assay in detecting emerging respiratory pathogens, as the continuous monitoring and updating of diagnostic tools are critical for responding to new public health threats.

## 5. Conclusions

In conclusion, the LabTurbo QuadAIO Common Flu Assay has excellent diagnostic accuracy and reliability in detecting multiple respiratory viruses, making it a powerful tool for clinical diagnostics. Its integration into routine clinical practice can enhance patient management, reduce the spread of infection, and improve healthcare outcomes. Further studies are warranted to confirm these findings and expand the applicability of this assay in various clinical settings.

## Figures and Tables

**Figure 1 diagnostics-14-02200-f001:**
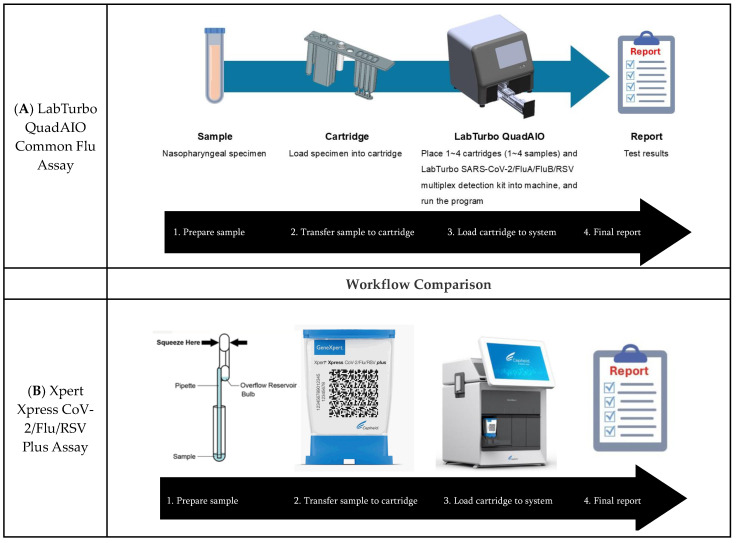
Workflow comparison of the LabTurbo QuadAIO Common Flu Assay and Xpert Xpress CoV-2/Flu/RSV Plus Assay. (**A**) LabTurbo QuadAIO system. (**B**) Xpert Xpress CoV-2/Flu/RSV Plus system.

**Figure 2 diagnostics-14-02200-f002:**
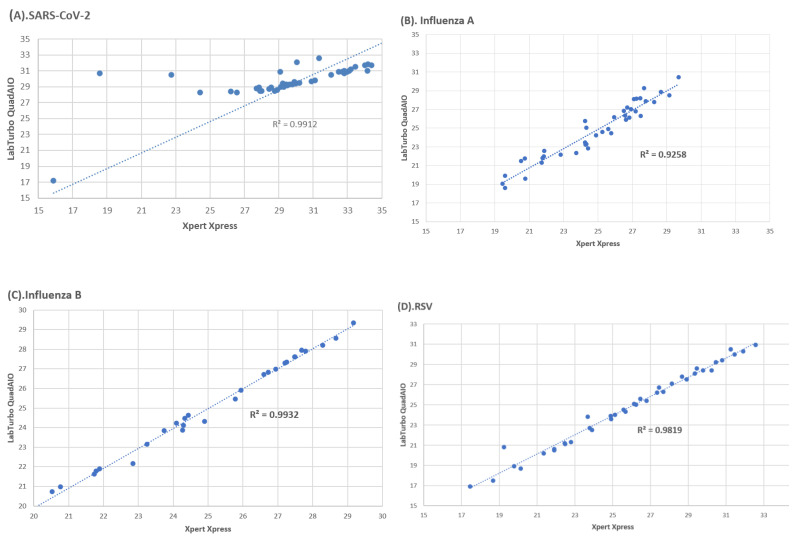
Correlation between Ct values for the Xpert Xpress CoV-2/Flu/RSV Plus (x-axis) and LabTurbo QuadAIO Assays (y-axis) for (**A**) SARS-CoV-2, (**B**) Influenza A virus, (**C**) Influenza B virus, and (**D**) RSV.

**Table 1 diagnostics-14-02200-t001:** Summary of performance metrics.

Virus	Positive Cases	Negative Cases	PPA (%)	NPA (%)
SARS-CoV-2	48	119	100	100
Influenza A	40	119	100	100
Influenza B	30	119	100	100
RSV	39	119	100	98.3

**Table 2 diagnostics-14-02200-t002:** SARS-CoV-2 clinical evaluation results between LabTurbo QuadAIO and Xpert Xpress Assays.

Sample ID	LabTurbo QuadAIO Assay Ct Value *	Xpert Xpress Assay Ct Value
TSGH-01	29.23	29.1
TSGH-02	30.05	32.1
TSGH-03	24.43	28.3
TSGH-04	26.20	28.4
TSGH-05	29.08	30.9
TSGH-06	28.00	28.5
TSGH-07	31.33	32.6
TSGH-08	36.26	30.9
TSGH-09	22.74	30.5
TSGH-10	18.58	30.7
TSGH-11	35.17	28.5
TSGH-12	29.23	29.4
TSGH-13	26.57	28.3
TSGH-14	15.88	17.2
TSGH-15	32.05	30.5
TSGH-16	31.10	29.8
TSGH-17	28.45	28.7
TSGH-18	29.30	29.0
TSGH-19	27.90	28.5
TSGH-20	33.00	30.9
TSGH-21	34.15	31.0
TSGH-22	32.80	30.7
TSGH-23	28.55	28.9
TSGH-24	29.45	29.3
TSGH-25	27.70	28.8
TSGH-26	29.90	29.6
TSGH-27	33.45	31.5
TSGH-28	34.20	31.8
TSGH-29	32.50	30.9
TSGH-30	30.90	29.7
TSGH-31	28.75	28.5
TSGH-32	29.65	29.3
TSGH-33	27.85	28.9
TSGH-34	29.50	29.2
TSGH-35	30.00	29.4
TSGH-36	32.80	31.0
TSGH-37	34.40	31.7
TSGH-38	33.10	31.0
TSGH-39	28.90	28.6
TSGH-40	30.20	29.5
TSGH-41	29.10	29.0
TSGH-42	27.80	28.8
TSGH-43	29.90	29.4
TSGH-44	33.20	31.2
TSGH-45	34.00	31.7
TSGH-46	32.70	30.9
TSGH-47	30.00	29.4
TSGH-48	29.80	29.3

* Ct Value: cycle threshold (Ct) value.

**Table 3 diagnostics-14-02200-t003:** Influenza A clinical evaluation results between LabTurbo QuadAIO and Xpert Xpress Assays.

Sample ID	LabTurbo QuadAIO Assay Ct Value *	Xpert Xpress Assay Ct Value
TSGH-101	27.48	28.17
TSGH-102	24.27	25.75
TSGH-103	28.66	28.83
TSGH-104	27.68	29.26
TSGH-105	24.9	24.19
TSGH-106	24.29	23.23
TSGH-107	27.79	27.84
TSGH-108	26.60	26.34
TSGH-109	25.78	24.43
TSGH-110	19.62	18.6
TSGH-111	19.62	19.88
TSGH-112	25.95	26.15
TSGH-113	24.43	22.83
TSGH-114	19.47	19.02
TSGH-115	25.60	24.9
TSGH-116	20.79	19.56
TSGH-117	26.83	26.12
TSGH-118	21.87	21.98
TSGH-119	29.71	30.44
TSGH-120	24.34	23.26
TSGH-121	26.52	26.82
TSGH-122	21.78	21.78
TSGH-123	27.25	28.11
TSGH-124	22.84	22.16
TSGH-125	21.88	22.56
TSGH-126	20.53	21.47
TSGH-127	29.17	28.47
TSGH-128	26.94	26.98
TSGH-129	24.33	25.0
TSGH-130	26.72	27.19
TSGH-131	28.29	27.77
TSGH-132	21.74	21.26
TSGH-133	20.77	21.73
TSGH-134	27.20	26.77
TSGH-135	23.74	22.34
TSGH-136	27.50	26.28
TSGH-137	24.25	23.44
TSGH-138	26.64	25.89
TSGH-139	25.27	24.56
TSGH-140	27.11	28.09

* Ct Value: cycle threshold (Ct) value.

**Table 4 diagnostics-14-02200-t004:** Influenza B clinical evaluation results between LabTurbo QuadAIO and Xpert Xpress Assays.

Sample ID	LabTurbo QuadAIO Assay Ct Value *	Xpert Xpress Assay Ct Value
TSGH-227	27.48	27.6
TSGH-228	24.27	23.8
TSGH-229	28.66	28.5
TSGH-230	27.68	27.9
TSGH-231	24.90	24.3
TSGH-232	24.29	24.1
TSGH-233	27.79	27.9
TSGH-234	26.60	26.7
TSGH-235	25.78	25.4
TSGH-236	19.62	19.2
TSGH-237	19.62	19.7
TSGH-238	25.95	25.9
TSGH-239	24.43	24.6
TSGH-240	23.25	23.1
TSGH-241	24.09	24.2
TSGH-242	21.78	21.7
TSGH-243	27.25	27.3
TSGH-244	22.84	22.1
TSGH-245	21.88	21.8
TSGH-246	20.53	20.7
TSGH-247	29.17	29.3
TSGH-248	26.94	26.9
TSGH-249	24.33	24.4
TSGH-250	26.72	26.8
TSGH-251	28.29	28.1
TSGH-252	21.74	21.6
TSGH-253	20.77	20.9
TSGH-254	27.20	27.2
TSGH-255	23.74	23.8
TSGH-256	27.50	27.6

* Ct Value: cycle threshold (Ct) value.

**Table 5 diagnostics-14-02200-t005:** RSV clinical evaluation results between LabTurbo QuadAIO and Xpert Xpress Assays.

Sample ID	LabTurbo QuadAIO Assay Ct Value *	Xpert Xpress Assay Ct Value
TSGH-257	21.34	20.2
TSGH-258	23.67	23.8
TSGH-259	19.78	18.9
TSGH-260	25.56	24.5
TSGH-261	31.89	30.3
TSGH-262	27.45	26.7
TSGH-263	24.90	23.6
TSGH-264	30.23	28.4
TSGH-265	22.48	21.1
TSGH-266	26.79	25.4
TSGH-267	28.90	27.5
TSGH-268	32.56	30.9
TSGH-269	19.23	20.8
TSGH-270	17.45	16.9
TSGH-271	23.78	22.7
TSGH-272	26.12	25.1
TSGH-273	29.45	28.6
TSGH-274	21.89	20.5
TSGH-275	27.34	26.2
TSGH-276	25.67	24.3
TSGH-277	31.45	30.0
TSGH-278	28.67	27.8
TSGH-279	24.89	23.9
TSGH-280	22.78	21.3
TSGH-281	26.45	25.6
TSGH-282	30.78	29.4
TSGH-283	20.12	18.7
TSGH-284	18.67	17.5
TSGH-285	29.34	28.1
TSGH-286	31.23	30.5
TSGH-287	23.89	22.5
TSGH-288	27.67	26.3
TSGH-289	25.12	24.0
TSGH-290	30.45	29.2
TSGH-291	21.89	20.6
TSGH-292	26.23	25.0
TSGH-293	29.78	28.4
TSGH-294	22.45	21.2
TSGH-295	28.12	27.1

* Ct Value: cycle threshold (Ct) value.

## Data Availability

The data presented in this study are available upon request from the corresponding author. The data are not publicly available because of the need to protect patient privacy.

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
