# Peer review of "Diagnostic Accuracy of the LabTurbo QuadAIO Common Flu Assay for Detecting Influenza A Virus, Influenza B Virus, RSV, and SARS-CoV-2"

_diagnostics, 2024, doi:10.3390/diagnostics14192200_

Round 1

Reviewer 1 Report

Comments and Suggestions for Authors

This work provided a detailed evaluation of the diagnostic accuracy of the LabTurbo QuadAIO Common Flu Assay compared to that of the Xpert Xpress CoV-2/Flu/RSV plus assay for detecting SARS-CoV-2, Influenza A, Influenza B, and respiratory syncytial virus (RSV), which can give some guidelines for doctors. There are issues the authors need to fix before this work can be published.

1. Could the authors provide some pictures to show the 'LabTurbo QuadAIO Common Flu Assay' and 'Xpert Xpress CoV-2/Flu/RSV plus assay' to let the readers have an intuitive idea about these assays? In addition, if possible, please also add the procedures of tests conducted on these assays/machines. 

2. It seems Table 2-6 are not necessary to be included in this manuscript since the date are also shown in Figure 1. Also, the readability of these tables is not so good. 

3. The discussion part is too long for such a short paper, I suggest the authors to make it brief. Frankly, the research content of this paper is not so much, and experimental design is simple. 

4. I suggest the authors to add the comparison of the test cost of these two assays, which is an important index of such assays. 

Author Response

  1. Could the authors provide some pictures to show the 'LabTurbo QuadAIO Common Flu Assay' and 'Xpert Xpress CoV-2/Flu/RSV plus assay' to let the readers have an intuitive idea about these assays? In addition, if possible, please also add the procedures of tests conducted on these assays/machines.

Response: Thank you for your suggestion. We included images of both the LabTurbo QuadAIO Common Flu Assay and the Xpert Xpress CoV-2/Flu/RSV plus assay to provide readers with a clearer visual understanding of these assays (Figure 1). Additionally, we described the testing procedures to help readers better comprehend the steps involved in line 108-112 and 118-122.

Xpert Xpress CoV-2/Flu/RSV plus assay

  1. It seems Table 2-6 are not necessary to be included in this manuscript since the date are also shown in Figure 1. Also, the readability of these tables is not so good.

Response: Thank you for your valuable feedback regarding the inclusion of Tables 2-6. To enhance the completeness of the data presented in the manuscript, we propose relocating Table 6 to the supplementary data section while retaining Tables 2-5 in the main manuscript.

  1. The discussion part is too long for such a short paper, I suggest the authors to make it brief. Frankly, the research content of this paper is not so much, and experimental design is simple.

Response: We have shortened the discussion section to focus on the key findings and their clinical implications in line 212 -216 . This will help streamline the content and maintain clarity, given the relatively simple experimental design of the study.

  1. I suggest the authors to add the comparison of the test cost of these two assays, which is an important index of such assays.

Response:  We appreciate your suggestion. We include a comparison of the testing costs for the two assays in the revised manuscript, as this is indeed an important metric for evaluating diagnostic tools. In line 239-243.

Reviewer 2 Report

Comments and Suggestions for Authors

This study aimed to evaluate the diagnostic accuracy of the LabTurbo QuadAIO Common Flu Assay in comparison to the Xpert Xpress CoV-2/Flu/RSV plus assay for detecting four respiratory pathogens: SARS-CoV-2, Influenza A, Influenza B, and respiratory syncytial virus (RSV). The results demonstrated that the LabTurbo QuadAIO Common Flu Assay provided comparable performance to the Xpert Xpress CoV-2/Flu/RSV plus assay.

In the introduction, the rationale for exploring alternative diagnostic assays beyond the Xpert Xpress CoV-2/Flu/RSV plus assay should be clearly explained. It is important to address the limitations of the Xpert assay, such as its cost, availability, or turnaround time, to justify the need for additional options like the LabTurbo QuadAIO assay. The introduction lacks sufficient information on the potential advantages of introducing the LabTurbo system, such as improvements in efficiency, cost-effectiveness, or broader accessibility in diagnosing respiratory viruses.

The methodology section requires refinement, particularly in describing the LabTurbo QuadAIO Common Flu Assay. Currently, the description reads more like promotional content rather than a technical explanation of the assay’s capabilities and clinical application. A more precise and neutral description of the assay's mechanisms, diagnostic targets, and workflow is important to lend credibility to the study.

Clarification is also needed regarding the routine use of both systems. Are both the LabTurbo QuadAIO and the Xpert Xpress assays commonly used for respiratory virus diagnosis, or is one preferred over the other in this hospital? According to the study, nasopharyngeal samples were first analyzed using the LabTurbo QuadAIO assay and then confirmed by the Xpert Xpress CoV-2/Flu/RSV plus assay, but it is unclear whether this reflects the standard diagnostic protocol in this setting. A more thorough explanation of the hospital's typical diagnostic procedures, including whether the retrospective samples used in the study followed standard practice, would strengthen the methodology section.

The primary findings focus on the agreement between the two assays. However, the inclusion of cycle threshold (Ct) values and correlation figures lacks sufficient justification. It is not explained why these results were included, nor is there any discussion on their relevance to the study. The Ct values should be contextualized, including what the established cut-off point for positivity is, and how they inform the overall diagnostic performance of the assays.

It remains unclear if the LabTurbo QuadAIO Common Flu Assay is regularly used in the hospital's routine laboratory work. If it is not, further explanation is needed to understand why this study was conducted and what potential role the LabTurbo assay could play in future routine diagnostics.

The discussion should focus on the potential advantages of the LabTurbo QuadAIO assay compared to other rapid molecular diagnostic tests. Key factors such as cost, turnaround time, throughput capacity, and overall accessibility should be explored in detail. Additionally, the discussion should address why the LabTurbo assay might be preferable to existing assays, considering practical factors like its scalability, ease of use, and potential for wider adoption in clinical settings. Currently, the study lacks sufficient discussion on why this assay should replace or complement other rapid molecular diagnostic tests.

Comments on the Quality of English Language

Some English language editing is needed.

Author Response

Comments and Suggestions for Authors

This study aimed to evaluate the diagnostic accuracy of the LabTurbo QuadAIO Common Flu Assay in comparison to the Xpert Xpress CoV-2/Flu/RSV plus assay for detecting four respiratory pathogens: SARS-CoV-2, Influenza A, Influenza B, and respiratory syncytial virus (RSV). The results demonstrated that the LabTurbo QuadAIO Common Flu Assay provided comparable performance to the Xpert Xpress CoV-2/Flu/RSV plus assay.

1.In the introduction, the rationale for exploring alternative diagnostic assays beyond the Xpert Xpress CoV-2/Flu/RSV plus assay should be clearly explained. It is important to address the limitations of the Xpert assay, such as its cost, availability, or turnaround time, to justify the need for additional options like the LabTurbo QuadAIO assay. The introduction lacks sufficient information on the potential advantages of introducing the LabTurbo system, such as improvements in efficiency, cost-effectiveness, or broader accessibility in diagnosing respiratory viruses.

Response: We revised the introduction to provide a clearer rationale for choosing the LabTurbo QuadAIO Common Flu Assay in line 71-82. Also, we included a discussion on the limitations of the Xpert assay, such as cost, availability, to justify the exploration of additional diagnostic options in line 239-243.

2.The methodology section requires refinement, particularly in describing the LabTurbo QuadAIO Common Flu Assay. Currently, the description reads more like promotional content rather than a technical explanation of the assay’s capabilities and clinical application. A more precise and neutral description of the assay's mechanisms, diagnostic targets, and workflow is important to lend credibility to the study.

Response: We revised the methodology section to present a more technical and neutral description of the LabTurbo QuadAIO Common Flu Assay line 108-112 and 118-122, including details on its diagnostic targets and workflow. This will lend more credibility to the study (also see Figure 1).

3.Clarification is also needed regarding the routine use of both systems. Are both the LabTurbo QuadAIO and the Xpert Xpress assays commonly used for respiratory virus diagnosis, or is one preferred over the other in this hospital? According to the study, nasopharyngeal samples were first analyzed using the LabTurbo QuadAIO assay and then confirmed by the Xpert Xpress CoV-2/Flu/RSV plus assay, but it is unclear whether this reflects the standard diagnostic protocol in this setting. A more thorough explanation of the hospital's typical diagnostic procedures, including whether the retrospective samples used in the study followed standard practice, would strengthen the methodology section.

Response: We included a more thorough explanation of the standard diagnostic protocols used in the hospital, clarifying and Xpert Xpress assays are routinely used as standard method and the whether both the LabTurbo QuadAIO assay could serve as an alternative diagnostic tool in line 93-99.

4.The primary findings focus on the agreement between the two assays. However, the inclusion of cycle threshold (Ct) values and correlation figures lacks sufficient justification. It is not explained why these results were included, nor is there any discussion on their relevance to the study. The Ct values should be contextualized, including what the established cut-off point for positivity is, and how they inform the overall diagnostic performance of the assays.

Response: Thank you for your comment on Ct values and correlation figures. We included these to show consistency between the assays, despite their qualitative nature. Comparing Ct values from LabTurbo QuadAIO and Xpert Xpress CoV-2/Flu/RSV Plus highlights their similar diagnostic performance.

The close Ct correlation indicates comparable sensitivity. Also, PPA and NPA values further demonstrate high concordance between assays, confirming LabTurbo QuadAIO's accuracy in detecting SARS-CoV-2, Influenza A/B, and RSV (Figure 1).

5.It remains unclear if the LabTurbo QuadAIO Common Flu Assay is regularly used in the hospital's routine laboratory work. If it is not, further explanation is needed to understand why this study was conducted and what potential role the LabTurbo assay could play in future routine diagnostics.

Response: We appreciate your suggestion. We included a comparison of the testing costs for the two assays in the revised manuscript, as this is indeed an important metric for evaluating diagnostic tools in line 239-243.

6.The discussion should focus on the potential advantages of the LabTurbo QuadAIO assay compared to other rapid molecular diagnostic tests. Key factors such as cost, turnaround time, throughput capacity, and overall accessibility should be explored in detail. Additionally, the discussion should address why the LabTurbo assay might be preferable to existing assays, considering practical factors like its scalability, ease of use, and potential for wider adoption in clinical settings. Currently, the study lacks sufficient discussion on why this assay should replace or complement other rapid molecular diagnostic tests.

Response: The Xpress CoV-2/Flu/RSV Plus assay costs around $68 per test, and the system itself is approximately $68,000. This high cost makes it less feasible for routine flu testing. In contrast, rapid COVID-19/Flu A/Flu B antigen test kits, which cost between $20 and $40 per kit and do not require an instrument, are much more affordable. The higher cost of the Xpress CoV-2/Flu/RSV Plus assay has hindered the widespread use of PCR tests for common flu, limiting their accessibility to the general population. We compared the cost between these two POCTs to discuss the LabTurbo QuadAIO assay might as an alternative diagnostic tool.

We revised the manuscript in line 239-243.

7.Comments on the Quality of English Language

Some English language editing is needed.

Response: We carefully edited the quality English grammar and wrong typo to make the manuscript more suitable for reader.

Reviewer 3 Report

Comments and Suggestions for Authors

The study by Jian and colleagues analyzes the performance of the LabTurbo QuadAIO Common Flu test for diagnosing respiratory tract infections caused by SARS-CoV-2, Influenza A and B and respiratory syncytial virus. The authors used the Xpert test as a reference.

The results show high diagnostic accuracy of the test for all viruses, presenting 100% PPA with the reference method and only two false-positives for RSV.

Although the manuscript is well written, some points should be clarified.

Authors must present the test supplier.

In the methodology, the authors describe that the study was prospective and retrospective. However, in the results it is not possible to assess which samples were used. For example, how many samples from inpatients and outpatients were included in the study?

Does table 1 show the results of the prospective or retrospective analysis?

“Nasopharyngeal swabs from the same patients were also analyzed.” (Line 93) Does this phrase refer to prospective or retrospective analysis?

Another example: “In our study, which included 30 positive and 100 negative cases”... These data refer to those presented in table 1.

Although in the text there is a description that the tables contain CT values, authors must add this information in the table title. (The titles shown are not self-explanatory).

Authors must review the text and reorganize the presentation of certain phrases such as:

“No evidence of cross-reactivity with other upper respiratory viruses was observed, as substantiated by the data delineated in Table 3.” (lines 190-191) Has this been observed only for Influenza A?

“These results indicated that the LabTurbo method has excellent sensitivity and specificity for successfully and accurately identifying all positive and negative cases (Tables 1 and 4).” (lines 197-198). Has this been observed only for Influenza B?

Table 6 can be presented as supplementary material.

Finally, the authors must explain why LabTurbo QuadAIO Common Flu test can be considered a POCT. (line 112).

Author Response

Comments and Suggestions for Authors

The study by Jian and colleagues analyzes the performance of the LabTurbo QuadAIO Common Flu test for diagnosing respiratory tract infections caused by SARS-CoV-2, Influenza A and B and respiratory syncytial virus. The authors used the Xpert test as a reference.

The results show high diagnostic accuracy of the test for all viruses, presenting 100% PPA with the reference method and only two false-positives for RSV.

Although the manuscript is well written, some points should be clarified.

1.Authors must present the test supplier.

Response: We presented the test supplier information for the LabTurbo QuadAIO Common Flu Assay and Xpert Xpress CoV-2/Flu/RSV Plus Assay used in this study to ensure the completeness of the manuscript.

2.In the methodology, the authors describe that the study was prospective and retrospective. However, in the results it is not possible to assess which samples were used. For example, how many samples from inpatients and outpatients were included in the study?

Does table 1 show the results of the prospective or retrospective analysis?

“Nasopharyngeal swabs from the same patients were also analyzed.” (Line 93) Does this phrase refer to prospective or retrospective analysis?

Another example: “In our study, which included 30 positive and 100 negative cases”... These data refer to those presented in table 1.

Response: The study is indeed retrospective, and we edited the methodology section to reflect this accurately. “prospective” will be removed to prevent confusion. We clarified the sources and numbers of samples used in the study (n=275 positive samples 156, negative samples=119) in the study design part. We also deleted some typo in the methodology section.

3.Although in the text there is a description that the tables contain CT values, authors must add this information in the table title. (The titles shown are not self-explanatory).

Response: We have revised the titles of all tables to clearly indicate that they contain cycle threshold (Ct) values. This adjustment ensures that the titles are self-explanatory and provide clear information about the content of each table.

4.Authors must review the text and reorganize the presentation of certain phrases such as:

“No evidence of cross-reactivity with other upper respiratory viruses was observed, as substantiated by the data delineated in Table 3.” (lines 190-191) Has this been observed only for Influenza A?

Response: We have deleted the typo in the revised manuscript.

“These results indicated that the LabTurbo method has excellent sensitivity and specificity for successfully and accurately identifying all positive and negative cases (Tables 1 and 4).” (lines 197-198). Has this been observed only for Influenza B?

Response: We have clarified the statement, “These results indicated that the LabTurbo method has excellent sensitivity and specificity for successfully and accurately identifying all positive and negative cases (Tables 1 and 4)”

Table 6 can be presented as supplementary material.

Response: We agree with your suggestion to present Table 6 as supplementary material. This change has been made, and Table 6 is now included in the supplementary section of the manuscript to streamline the main text and focus on the primary results.

Finally, the authors must explain why LabTurbo QuadAIO Common Flu test can be considered a POCT. (line 112).

Response: We include a detailed explanation of why the LabTurbo QuadAIO Common Flu Assay is considered a point-of-care test (POCT) in discussion and introduction part, highlighting its rapid diagnostic capabilities, ease of use, and suitability for various clinical settings.

Round 2

Reviewer 2 Report

Comments and Suggestions for Authors

The author has made all the necessary adjustments.